# Study on the Liquid Cooling Method of Longitudinal Flow through Cell Gaps Applied to Cylindrical Close-Packed Battery

**Wei Li** [1] , **Wei Shi** [2], **Shusheng Xiong** [2,3,4,5,*], **Hai Huang** [2] **and Guodong Chen** [2]

1    School of Mechanical Engineering, Zhejiang University, Hangzhou 310030, China; 11725081@zju.edu.cn
2    College of Energy Engineering, Zhejiang University, Hangzhou 310014, China; shiw@zju.edu.cn (W.S.);
     22227001@zju.edu.cn (H.H.); 0920471@zju.edu.cn (G.C.)
3    Key Laboratory of Clean Energy and Carbon Neutrality of Zhejiang Province, Hangzhou 310014, China
4    Jiaxing Research Institute, Zhejiang University, Jiaxing 314016, China
5    Longquan Industrial Innovation Research Institute, Lishui 323700, China
*    Correspondence: xiongss@zju.edu.cn

**Abstract:** The increasing popularity of electric vehicles presents both opportunities and challenges for the advancement of lithium battery technology. A new longitudinal-flow heat dissipation theory for cylindrical batteries is proposed in order to increase the energy density and uniform temperature performance of cylindrical lithium-ion battery packs while also shrinking their size by roughly 10%. First, a genetic algorithm is used to identify a single cell's thermal properties. Based on this, modeling and simulation are used to examine the thermal properties of the longitudinal-flow-cooled battery pack. It is found that the best coolant flow scheme has one inlet and one outlet from the end face, taking into account the cooling effect of the battery pack and engineering viability. Lastly, thermal dummy cells (TDCs) are used to conduct a validation test of the liquid cooling strategy. Additionally, the simulation and test results demonstrate that the liquid cooling solution can restrict the battery pack's maximum temperature rise under the static conditions of a continuous, high-current discharge at a rate of 3C to 20 °C and under the dynamic conditions of the New European Driving Cycle (NEDC) to 2 °C. In applications where the space requirements for the battery pack are quite strict, the longitudinal-flow cooling method has some advantages.

**Keywords:** liquid cooling; longitudinal flow; cylindrical cell; dummy cell





## 1. Introduction

The transportation sector has started to electrify in response to the crises of fossil fuels and global air pollution. Rail transportation has pioneered the popularization of electric drives. The transition from gasoline-powered to electric vehicles has begun in the road transportation sector as a result. Plans to outlaw the selling of fuel-powered vehicles have been put forth by nations all over the world. By the end of 2022, China has already achieved their target of 20% penetration of new-energy vehicles by 2025 ahead of schedule. While hydrogen fuel cell technology is gaining traction as another new energy supply solution for new-energy cars, pure electric vehicles with lithium-ion battery packs and drive motors as the power system are the industry mainstream [1]. The lithium-ion battery is the essential part of any car's energy supply system, whether it be a fuel cell vehicle or a pure electric vehicle.

New-energy vehicles have benefits over conventional internal combustion engine vehicles, such as high efficiency and cleanliness. However, there are also some distinct drawbacks, the most notable of which are the stringent criteria for the environmental temperature during use. All power sources have specific temperature ranges, whether they are drive motors, fuel cells, or power lithium-ion batteries. The normal operating temperature range for a lithium-ion battery is between −10 °C and +50 °C. Considering their performance and durability, vehicle lithium-ion power batteries are sometimes restricted

to a narrower operating temperature range, which is between +20 °C and +40 °C [2–5]. Between +45 °C and +60 °C is the ideal operating temperature for fuel cells [6]. The operating temperature range for the drive motor is greater, with a maximum temperature of +130 °C [7,8]. During the charging and discharging process, lithium-ion batteries inevitably produce Joule heat and chemical reaction heat. The battery temperature rise brought on by the accumulation of heat cannot be disregarded because it will negatively impact the battery's energy efficiency, safety, and life [9]. Therefore, the powertrain of new-energy vehicles needs to be developed with a dependable battery thermal management system (BTMS) that can adapt to both cold and hot climates for global adoption.

As early as the beginning of 2000, Pesaran et al. [10] recognized the importance of thermal management of EV and HEV batteries. Over a period of more than 20 years, researchers have created several cooling techniques for managing the temperature of batteries. These techniques include gas, liquid, phase-change materials (PCM), heat pipes (HP), peltier, and various combinations of these methods [11]. It has been discovered that the battery thermal management system must consider not only heat dissipation requirements but also the vehicle's thermal management, preheating the battery in low-temperature conditions, power consumption, volume, temperature uniformity, and other design specifications [12]. While combining different cooling methods can effectively achieve the cooling function, there is also high potential in optimizing single cooling methods such as liquid cooling and air cooling. Liquid cooling has clear advantages over air cooling and is the dominant method used in commercial applications [13]. Passive cooling methods, such as PCM and HP, have gained widespread attention due to their exceptional performance [14]. PCM has various designs, including cooling for lithium batteries, electronic components, and photovoltaic panels [15,16]. Despite its benefits, PCM has some drawbacks such as low thermal conductivity, which makes it susceptible to failure due to heat accumulation, and low volume density. On the other hand, the HP material has a high manufacturing cost, is complex to install, and is difficult to maintain. Despite numerous optimization schemes proposed in [17], practical commercial applications are still a long way off [18]. Currently, liquid cooling is the most widely used solution for managing battery temperatures due to its technical effectiveness, ability to dissipate heat, and cost-effectiveness.

Transverse flow and series connection are mostly employed for the heat dissipation of cylindrical battery packs that are either liquid-cooled or air-cooled. Although the design of the flow channel for the transverse-flow heat dissipation method is relatively straightforward, the fluid transverse-flow channel must pass through the cells, necessitating the maintenance of a specific gap between each cell. This requires additional space to be occupied and is inimical to increasing the energy density of the battery pack. The surface temperatures of each individual cell in a series heat dissipation battery pack are also out of balance due to the large temperature difference along the direction of fluid flow, which reduces capacity and shortens battery pack life [19]. Individual studies have proposed some forms of longitudinal-flow heat dissipation for cylindrical cells, which can improve the uniform temperature performance of the battery pack; however, these solutions still require a certain gap between individual cells and do not effectively improve the space utilization of the battery pack.

The longitudinal-flow heat dissipation proposed in this paper is a form of convective heat transfer in which the heat transfer fluid flows axially along the cell in the apertures formed by cylindrical cells in a tightly arranged group and forms a convective heat transfer between the cell and the cylindrical surface. The solution has three features: first, the arrangement characteristics of cylindrical lithium-ion batteries mean they can be used as coolant flow channels with the help of their arranged pores, so that the cells can be closely arranged, effectively reducing the size of the battery pack and improving the energy density of the cylindrical battery pack; second, for each cell, the heat transfer area is the same and the inlet temperature of the heat transfer fluid is also the same, so the battery pack has better uniform temperature performance; third, the heat transfer fluid flows along the axial

direction of the cell, and the entire side of the cell can be used for heat dissipation, thus obtaining a higher heat transfer efficiency. The research contribution of this program is reflected in two aspects.

1. Improved energy density of cylindrical battery packs

Compared with natural cooling and forced air cooling, the liquid flow method of cooling batteries is widely used because of its high efficiency, reliability, and other characteristics [20]. However, the extra weight of the coolant itself, the coolant flow channels, and the sealing elements used to avoid coolant leakage increase the manufacturing cost of the coolant circulation system, which is a disadvantage of the liquid cooling method [21]. For liquid cooling of cylindrical cells, all methods proposed or in use today require a certain gap between all the individual cells in the diameter direction to allow a coolant flow path to pass through, which undoubtedly increases the size of the battery pack and reduces its volumetric energy density. The longitudinal-flow heat dissipation method proposed in this paper aims to eliminate this gap by exploiting the arrangement characteristics of cylindrical Li-ion cells and using their arranged pores as coolant flow paths to pass cooling media along the axial direction of the cells for heat dissipation. This approach allows the cylindrical cells to be closely arranged, which not only reduces the material and weight of the coolant flow channel but also increases the volumetric energy density of the whole battery pack.

2. Improved uniform temperature performance of cylindrical battery packs

The conventional cooling method for cylindrical cells is transverse-flow series cooling. The cooling medium flows in the direction of the cell's diameter and is in direct or indirect contact with the sides of the individual cells. In the direction of flow of the cooling medium, heat is exchanged with each individual cell in turn. Therefore, the closer to the outlet, the higher the temperature of the cooling medium, and the lower the heat exchange rate with the surface of the cell. The worse the heat dissipation effect, the higher the surface temperature of the cell. This is the biggest disadvantage of the transverse-flow cooling method. In contrast, with the longitudinal-flow cooling method, each cell can receive an even temperature coolant, and there is no such problem.

## 2. Method and Experiment

### 2.1. Simulation

#### 2.1.1. Longitudinal-Flow Geometry Model

Figure 1 shows the longitudinal-flow liquid cooling model, which consists of seven cylindrical cells closely arranged in green as shown in the figure. In the test, in order to prevent the cooling water from making contact with the cells and leakage causing test failures or even safety problems, a layer of sidewalls with a thickness of 0.5 mm is applied to the cells to completely isolate the cells from the cooling liquid. Therefore, the battery is separated in the radial direction by two layers of sidewalls, the thickness of which is 1 mm. In the actual project, insulated coolant should be employed, and the clearance sidewalls can be eliminated, so that, through a more advanced metal-welding process, the thickness of the sidewalls can be greatly reduced; alternatively, direct contact should be used.

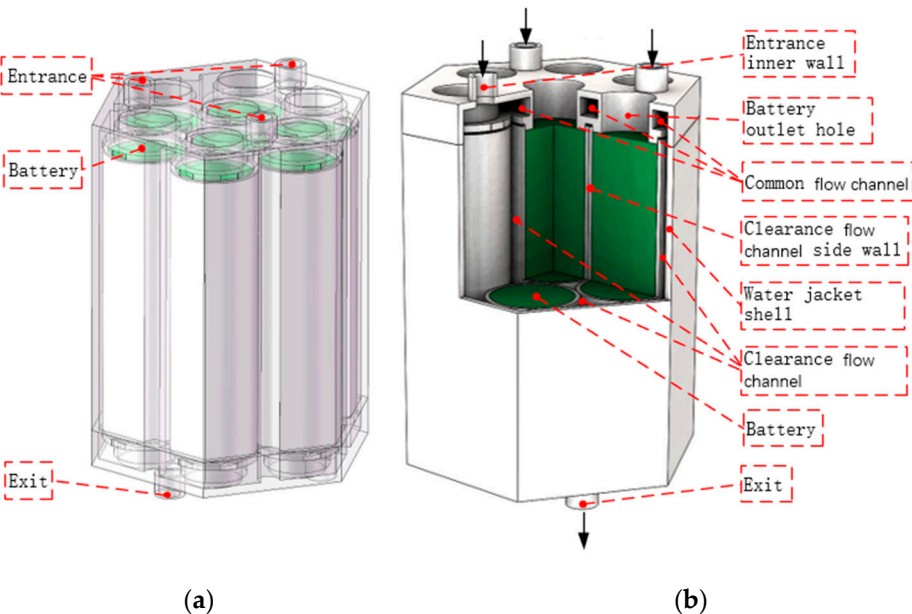

(**a**)　　　　　　　　　　　　　　　　　　　　　　　(**b**)

**Figure 1.** Schematic diagram of the battery pack structure: (**a**) translucent view; (**b**) 1/8 sectional view.

### 2.1.2. Model Calculation Domain

Based on the above structural design scheme, the geometric model for simulation is established in the multi-physics field simulation software COMSOL based on the three-dimensional model of the battery pack shown in Figure 1. Three physical fields, solid and fluid heat transfer, turbulent flow, and lumped cell, are added to the software and applied to the water jacket shell, coolant, and cell, respectively, to verify the thermal characteristics of the battery pack. In addition, two multi-physical field interfaces are coupled through non-isothermal flow and electrochemical heat.

It is worth noting that since the outer sidewalls of the two cells are tangential in the actual model, singularities will occur during COMSOL meshing, resulting in mesh generation failure. To avoid this problem, the diameter of the total cell is set to 17 mm in the simulation model, while the thin layer property is added to its sidewalls, and the thickness of the thin layer is set to 0.5 mm. In addition, an inner wall, as shown in Figure 1, is set between the sidewalls of the simulated cell to isolate the flow between the gap flow channels.

### 2.1.3. Genetic Algorithm

To simulate with COMSOL, it is essential to utilize the thermal conductivity coefficients and convective heat transfer coefficients of the actual cells. It should be noted that the thermal conductivity of a cell differs in the axial and radial directions. To achieve more precise thermal coefficients, the output is based on the temperature of a single cell during discharge, while the inputs include entropy coefficients, cell terminal voltage, and open-circuit voltage. A genetic algorithm is utilized to identify the thermophysical parameters of the cell. For the thesis simulation, a lumped parameter model is used to represent the battery. The battery is treated as a uniformly heat-generating object that produces heat from within and conducts it outward. The heat is then transferred to the coolant through the battery's surface and carried out of the battery pack by the coolant. During this process, the accuracy of the simulation results is directly determined by the battery's thermal conductivity and surface heat transfer coefficient, assuming the battery heat generation power is correct.

2.1.4. Control Equation and Related Parameters

Heat Generation Mechanism of Battery

The cell model used in this paper is provided by the lumped cell interface in the simulation software, and its electrochemical heat generation equation is shown in Equation (1).

$$Q_h = \left( \eta_{IR} + \eta_{act} + T_{cell} \frac{\partial E_{OCV}(SOC, T_{cell})}{\partial T} \right) I_{cell} + Q_{mix} \tag{1}$$

where $Q_h$ is the heat generation power of the battery, in W; $\eta_{IR}$ is ohmic overvoltage, in V; $\eta_{act}$ is the activation overpotential, in V; $Q_{mix}$ is the heat of mixing, in W; and $I_{cell}$ is the charge and discharge current of the battery, in A. The third item in the brackets is the product of the open-circuit voltage temperature coefficient of the battery and the average temperature of the battery, indicating the heat of the chemical reaction of the battery, which can be expressed as heat absorption or heat release. In calculation, activation overpotential and mixing heat are often neglected [22], so electrochemical heat is mainly generated by ohmic overpotential and chemical reaction heat. The simplified heat generation formula is shown in Formula (2):

$$Q_h = I_{cell} \left( \eta_{IR} + T_{cell} \frac{\partial E_{OCV}(SOC, T)}{\partial T_{cell}} \right) \tag{2}$$

The expression of ohmic overpotential is shown in Formulas (3) and (4).

$$\eta_{IR} = \eta_{IR,1C} \frac{I_{cell}}{I_{1C}} \tag{3}$$

$$\eta_{IR,1C} = \eta_{1C} Arrh\left(Ea_{\eta_{1C}}, T_{cell}\right) \tag{4}$$

In these equations, $\eta_{IR,1C}$ is 1C ohmic overvoltage, in V; $I_{1C}$ represents the current of the battery at 1C magnification, in A; $\eta_{1C}$ is a constant representing the 1C ohmic overvoltage at the reference temperature, in V; and $Ea_{\eta_{1C}}$ is a constant indicating the activation energy of the battery, in J/mol. $T_{cell}$ is the average temperature of the battery, in K. In the formula, the function *Arrh* is the Arrhenius equation, which represents the relationship between the chemical reaction rate constant and temperature. The exponential form is used here, and the formula is as follows:

$$\text{Arrh}(x, y) = e^{\frac{Ea}{R} \left( \frac{1}{T_{cell}} - \frac{1}{T_0} \right)} \tag{5}$$

$R$ is the molar gas constant, and its value is 8.314 J/(mol·K); $Ea$ represents the activation energy of the battery, and the calculation process is substituted into $Ea_{\eta1C}$. $T_0$ is the reference temperature, which is 293.15 K in this paper.

Flow and Heat Transfer Equation

The differential equations for the heat transfer of solids and fluids are shown in Equations (6), (8), and (9).

$$\rho c_p \frac{\partial T}{\partial t} = \frac{1}{r} \frac{\partial}{\partial r} \left( k_r r \frac{\partial T}{\partial r} \right) + \frac{1}{r^2} \frac{\partial}{\partial \varphi} \left( k_\varphi \frac{\partial T}{\partial \varphi} \right) + \frac{\partial}{\partial z} \left( k_z \frac{\partial T}{\partial z} \right) + Q \tag{6}$$

$$\rho c_P \frac{\partial T}{\partial t} + \rho c_P u \cdot \nabla T + \nabla \cdot q = 0 \tag{7}$$

$$q = -k \nabla T \tag{8}$$

Equation (6) is the differential equation for the thermal conductivity of the solid cell in the cylindrical coordinate system. Equation (7) is the differential equation for the heat transfer of the fluid. Here, $\rho$ is the density in kg/m$^3$; $c_P$ is the specific constant pressure

heat capacity, in J/(kg·K); $T$ is the temperature, in K; $t$ is the time, in seconds; $u$ is the vector of fluid velocity, in m/s; $q$ is the heat flux in W/m$^2$; $Q$ is the heat generation power of the heat source, in W; and $k, k_r, k_\varphi, k_z$ represent the thermal conductivity, in W/(m·K).

The temperature and velocity field distributions can be derived from this equation, where the $Q$ represents the battery heat generation power, so as to realize the coupling of electrochemical heat and fluid heat transfer.

The actual Reynolds number of coolant flow in the water jacket of the cell is calculated by the following equation:

$$R_e = \frac{u_a D}{v} \tag{9}$$

where $R_e$ is the Reynolds number of the actual flow; $u_a$ is the average flow velocity, in m/s; $D$ is the equivalent diameter, in m; $v$ is the kinematic viscosity of the fluid, in m$^2$/s; and the kinematic viscosity of water at normal temperature is $1.006 \times 10^{-6}$ m$^2$/s. The equivalent diameter $D$ of the gap is calculated by the following formula:

$$D = 4\frac{A}{\chi} \tag{10}$$

where $A$ is the effective sectional area, in m$^2$, and $\chi$ is the wet circumference, in m.

According to the three-dimensional structure model shown in Figure 1, the effective sectional areas of the two clearances are calculated to be 15.52 mm$^2$ and 42.43 mm$^2$, respectively. Their corresponding wet circumferences are 21.43 mm and 40.32 mm. The average velocity is calculated at 4 m/s, and their maximum Reynolds numbers are 11,518 and 16,774, which are close to or more than the empirical value of the upper Reynolds number 13,800 [23]. Therefore, K-ε turbulence model is adopted in the simulation mode.

In the COMSOL lumped battery numerical model, there are two important parameters to be configured: one is the open-circuit voltage and state of charge (OCV-SOC) characteristic curve, and the other is the temperature coefficient of the battery open-circuit voltage. These two parameters are measured by specific test aforehand.

### 2.1.5. Initial and Boundary Conditions

The initial and boundary conditions of the model used are shown in Table 1. The other parameters used in the model are shown in Table 2.

**Table 1.** Initial and boundary conditions.

| Number | Conditions |
|:---:|:---:|
| 1 | The initial temperature and reference temperature of the model and environment are set as 20 °C |
| 2 | The outer wall of the water jacket in contact with the air is set as an adiabatic surface |
| 3 | The heat convection between the two end surfaces of the cell and the heat conduction of the wires are set to a constant heat flux value |
| 4 | The contact thermal resistance of the battery sidewalls and coolant water jacket is ignored |
| 5 | The thermal property parameters of the materials used are treated as constant values |
| 6 | The coolant is set as water, the fluid is considered incompressible, and the gravity of water is ignored |
| 7 | The fluid inlet is set according to the normal inflow speed, and the outlet is set to zero pressure |

**Table 2.** Parameters used in the model.

| Condition | Value | Unit |
|---|---|---|
| Initial temperature | 20.0 | °C |
| Equivalent heat flux of battery end face | 247.19 | $W/(m^2 \cdot K)$ |
| Radial thermal conductivity of battery | 2.37 | $W/(m \cdot K)$ |
| Axial thermal conductivity of battery | 28.27 | $W/(m \cdot K)$ |
| Battery capacity | 3.25 | Ah |
| Specific heat capacity of battery at constant pressure | 2482.3 | $J/(kg \cdot K)$ |
| Battery density | 2841.60 | $kg/m^3$ |

Water is a commonly used coolant in various applications such as machinery, electrical products, and internal combustion engines. The most frequently used coolants in BTMSs include water and a solution of ethylene glycol and water [13,24]. The coolants used in BTMSs include water, ethylene glycol water solution, oil, nanofluid, and liquid metal. According to a paper [25], nanofluid and liquid metal are superior coolants in terms of heat dissipation performance compared to water in numerical simulation. Oil is often used as a coolant for the direct cooling method, which has a unique advantage. Ethylene glycol water solution reduces the cooling ability of water, but the degree of reduction depends on the concentration of ethylene glycol in mass. The most commonly used antifreeze is a mass concentration of 50% ethylene glycol. However, a comparison in another paper [26] showed that the cooling effects of ethylene glycol water solution and pure water had a maximum difference in temperature of less than 2 °C, and the maximum difference in the temperature difference was less than 1 °C. Therefore, using water as the coolant for the validation of the cooling method in this thesis is reasonable.

### 2.1.6. Grid Validity Verification

As shown in Table 3, six cases are selected to verify the effect of meshing on the cell simulation temperature. The meshing strategy is based on five modes predefined by the software, which are super coarsening, relatively coarsening, coarsening, conventional, and refinement. All simulation conditions are kept consistent except for the mesh division.

**Table 3.** Grid division patterns and quantity.

| Condition | Grid 1 | Grid 2 | Grid 3 | Grid 4 | Grid 5 | Grid 6 |
|---|---|---|---|---|---|---|
| Predefined mode | Super coarsening | Relatively coarsening | Coarsening | Conventional | Refinement | Relative refinement |
| Number of domain units | 42,756 | 73,005 | 123,990 | 229,310 | 612,397 | 1,310,894 |
| Number of boundary elements | 8025 | 12,066 | 17,408 | 25,652 | 46,129 | 71,452 |
| Number of side cells | 1314 | 1663 | 2081 | 2532 | 3673 | 4598 |

One of the batteries is selected as the research object, its average temperature is calculated under the above five meshing conditions, and the simulation results are obtained as shown in Figure 2a. It can be seen from the figure that the greater the number of grids, the lower the average temperature. The maximum temperature difference of the five cases is within ±0.5 °C. The average temperature shown in the main axis of Figure 2b is the value when the Depth of Discharge (DOD) in Figure 2a is 50%, and the time shown in the secondary axis is the number of hours consumed in the complete discharge simulation of the battery pack. As can be seen from the graph, the time required for the simulation increases proportionally with the number of grids, while the difference in the average temperature after the refinement of grid 5 is small and can be ignored. In order to balance simulation accuracy and efficiency, grid 4 is selected for the simulation model used in this paper.

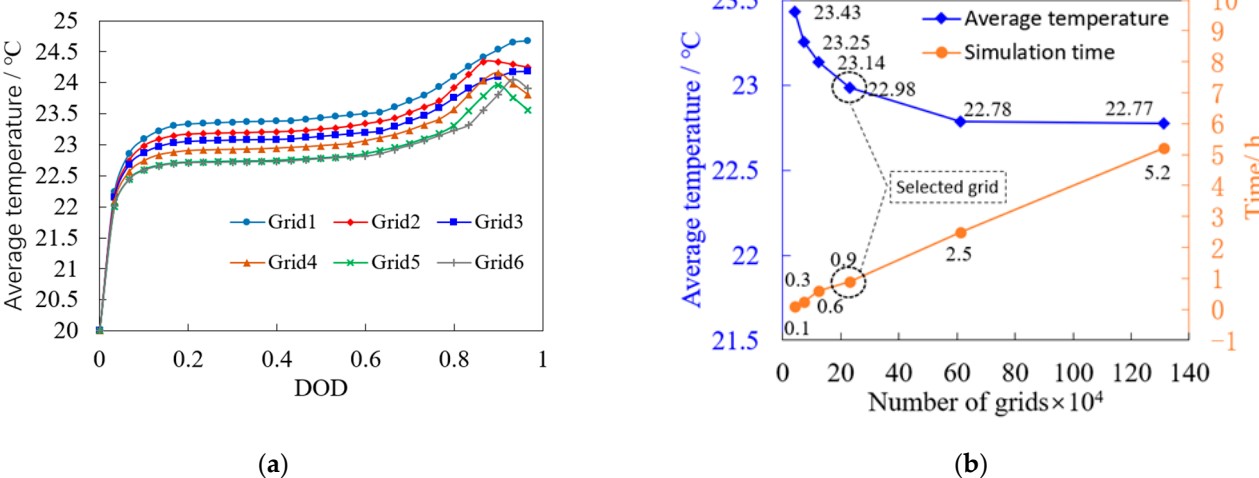

(**a**)                                                                                              (**b**)

**Figure 2.** Comparison of average temperature and simulation time for five meshing cases: (**a**) average temperature of the battery during discharge; (**b**) average temperature and simulation time for cases with different number of meshes.

*2.2. Experiment*

2.2.1. TDC-Based Pack Principle

In previous publications by our team, we confirmed the possibility of creating a battery thermal management system (BTMS) using TDC in place of an actual battery [27]. In this thesis, Figures 3 and 4 present the TDC battery's physical structure and the experimental assembly structure with the TDC battery, respectively. To simulate the behavior of an actual battery, TDC uses electric heating wire to directly heat the coiling filler. This ensures that the TDC has a similar or greater thermal power compared to the battery. Additionally, the TDC's axial and radial thermal conductivity, as well as convective heat transfer coefficient, are comparable to the actual battery. By measuring the thermal power of the battery during charging and discharging, TDC can simulate the battery's thermal behavior accurately. Using this principle, the TDC can effectively simulate the thermal behavior of batteries across a wide range of operating rates. Even in the event of thermal runaway, this method provides a much safer and more dependable option.

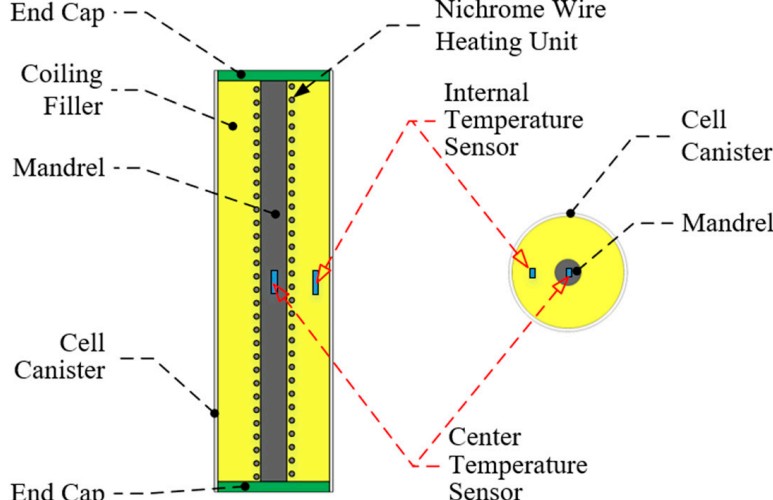

**Figure 3.** Distribution diagram of TDC temperature sensor in the test.

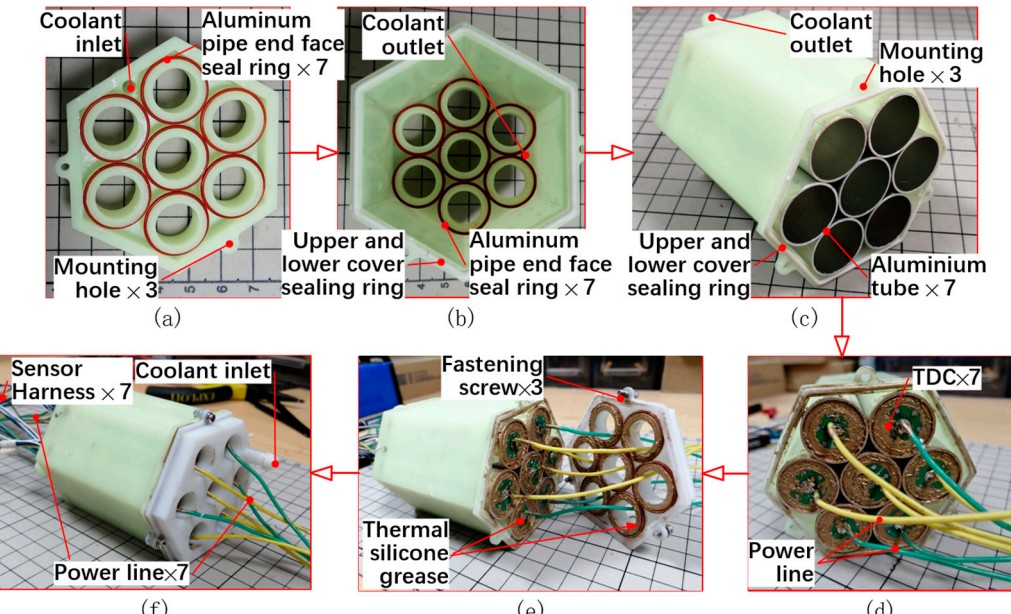

**Figure 4.** Assembly process and structural principle of battery pack prototype with Gap flow channel: (**a**) after the upper cover has the seal attached; (**b**) after the lower cover has the seal attached; (**c**) after the lower cover is loaded into the aluminum tube; (**d**) after the lower cover is loaded with the TDC; (**e**) before the upper and lower covers are combined; (**f**) after the upper and lower covers are tightened.

### TDC Structure

The battery pack used in this experiment consists of seven thermal dummy cells (TDCs) researched by the authors, the details of which can be found in our previous publication [27]. A TDC was used instead of the real cell for thermal power simulation, and both internal and core temperatures were measured to obtain the extremes and variation curves of temperature for different cell operating conditions and coolant flow rates. The internal structure and principle of the used TDC are basically the same as the simulation model. Here, the core of the TDC is a hollow alumina ceramic rod with 4 mm outer diameter and 2 mm inner diameter to facilitate the arrangement of a temperature sensor in the center.

Two temperature sensors are embedded inside each TDC. One is placed inside the mandrel, called the central temperature sensor, and the other is placed at the axial midpoint inside the winding fill, which is separated from the outer shell by an approximately 3 mm thick thermally conductive silicone fill, called the internal temperature sensor. The TDC is similar to the battery in that the highest temperature of its sidewall is at the midpoint, so the reading of the internal temperature sensor is representative of the average temperature of the actual cell. Since the center temperature sensor is very close to the heating unit, the temperature response at this point is fast and reflects the amount of heating power, representing the maximum temperature inside the TDC.

Due to a production error, there is a certain deviation in the resistance of the heating units of the 7 TDCs, and their resistance values and average values are shown in Table 4. It can be seen that the maximum resistance deviation between them is within 3.8%, and under the same supply voltage, the corresponding heating power will also have a certain deviation, albeit not too large of one.

**Table 4.** Resistance value and average value of heating unit in each TDC.

| TDC No. | 1 | 2 | 3 | 4 | 5 | 6 | 7 |
|---|---|---|---|---|---|---|---|
| Resistance value/$\Omega$ | 14.74 | 14.84 | 14.32 | 14.87 | 14.60 | 14.53 | 14.86 |
| Average value of resistance/$\Omega$ | | | | 14.68 | | | |

Design of TDC Battery Pack and Liquid-Cooled Heat Exchange Structure

According to the structure model in Figure 1, a set of battery-pack-cooling water jacket prototypes were made using 3D printing technology. The resulting real object is shown in Figure 4. The assembly process and internal structure details of the battery pack prototype are described in the figure.

The prototype has two difficulties. One is sealing and waterproofing. This paper adopts O-type silicone seal with wire diameter 1.2 mm to seal between the aluminum tube and the upper and lower cover, and it adopts foam silicone seal with wire diameter 1.0 mm to seal between the upper and lower cover. The second is thermal conductivity. This paper adds thermal conductive silicone grease (QM950; Shanghai Zihong Electronics Co., Shanghai, China) between the TDC and the aluminum tube to reduce the contact thermal resistance. The thermal conductivity is greater than 4 W/(m·K) according to the product specification provided by the manufacturer. In particular, the battery pack prototype used in this test was assembled with aluminum tubes with a wall thickness of 0.5 mm, resulting in a radial cell clearance of 1 mm. In practice, this clearance can be reduced to 0.5 mm or less by manufacturing processes such as custom aluminum profiles and brazing. The temperature measurement system used in this experiment can be found in the aforementioned literature [28].

### 2.2.2. Test System and Method
Test System Composition

In order to verify the feasibility of the new liquid cooling scheme based on the above battery pack prototype, a dedicated test bench needs to be built to establish the coolant circulation, signal sampling, and power supply [29]. In this paper, a test bench scheme is designed, as shown in Figure 5.

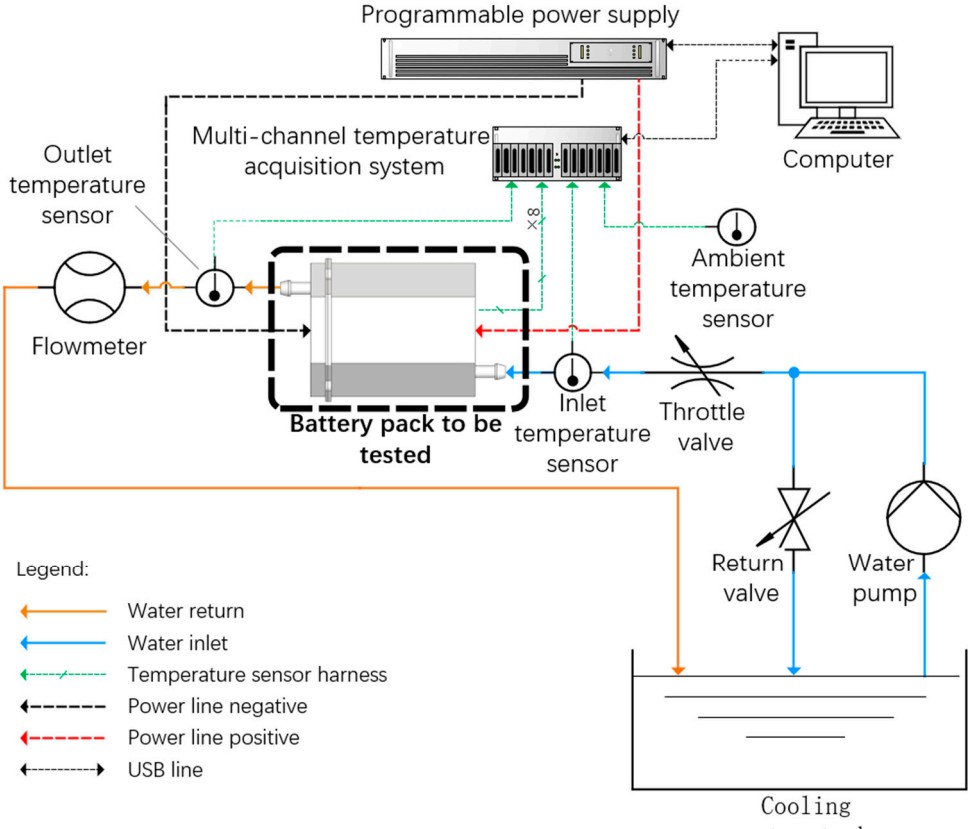

**Figure 5.** Schematic diagram of test bench.

The test bench consists of a water pump, two regulating valves, a flow meter, and a water tank to form a coolant circulation channel. A programmable power supply (IT6516D; Itech Electronics Co, Ltd., Nanjing, China) with an output power of 1.8 kW supplies power to seven parallel TDCs, and a multi-channel temperature acquisition system measures the temperature of each TDC, the coolant inlet and outlet, and the ambient air. The flow meter used for the test was a gear-type liquid flow meter with a range of 0.1–10 L/min and a measurement accuracy of ±0.5%. The pump used has a head of 2.8 m and a maximum flow rate of 60 L/min. In order to avoid leakage of coolant into the TDC, which may cause temperature measurement errors, industrial deionized distilled water is used as the coolant in this test.

The pump used in the test is a submersible pump, which needs to be submerged in the water tank for use. The battery pack in the test is supported by two bases and placed vertically. Its water inlet is at the bottom, and the water outlet is at the top, so that the water jacket of the battery pack can be filled with coolant even under the test conditions of low flow rate. Since the cooling water for the test is recycled and the test bench is not equipped with heat sink, the water tank should be as large as possible to ensure constant water temperature. The water tank for this experiment is a 50 L plastic tank. In the test, the tank contains 30 L of deionized water. As the water pump used in the test bench is a fixed-speed pump, in order to achieve the regulation of the cooling water flow, a precision-adjustable throttle valve is employed, and a return valve is also used, so as to avoid blockage of the water pump in the case of the throttle valve being closed, resulting in motor overload failure. In the beginning of the test, the return valve is kept in the fully open state, and it is then gradually closed when the cooling water flow needs to be further increased.

Test Procedure

The experiments described in the following sections of this paper are conducted as follows:

(1) Establish cooling water circulation to bring the flow rate up to the predetermined value.
(2) Start the temperature acquisition system, observe, and wait for the coolant inlet and outlet temperatures to become essentially constant.
(3) Import the predefined output voltage sequence into the programmable power supply.
(4) Clear the temperature acquisition system's historical data and start a new acquisition while starting the power supply output.
(5) Wait for the programmable power supply to complete the voltage sequence output, and then end the temperature sampling and save the data.

## 3. Results

### 3.1. Analysis of Simulation Results

3.1.1. Comparison of Cooling Performance with Transverse-Flow Battery Pack

The geometric model of the longitudinal flow is shown in Figure 1. According to the practical working condition of transverse-flow cooling of the battery, on the basis of the longitudinal-flow cooling model, the axis distance of the cell is increased to 22 mm and the coolant exposure height is set to 32.5 mm. Orderly transverse flow of coolant is achieved by setting a partition. Keeping the battery heat power, inlet and outlet area, and flow rate constant, the following curves are obtained by simulation:

Figure 6a compares the maximum and minimum temperatures of the battery pack under longitudinal-flow and transverse-flow cooling conditions. Figure 6b compares the maximum temperature difference of different cells in the battery pack under different cooling conditions, and the figure specifies the maximum and minimum temperature difference of cells, including the maximum temperature difference between longitudinal flow and transverse flow, as well as the overall maximum temperature difference. As can be seen from the figure, the temperature rise of the battery pack after cooling by transverse flow and longitudinal flow is almost the same under the simulation conditions, but the

longitudinal flow can play a positive role in reducing the temperature difference in both individual cells and the battery pack.

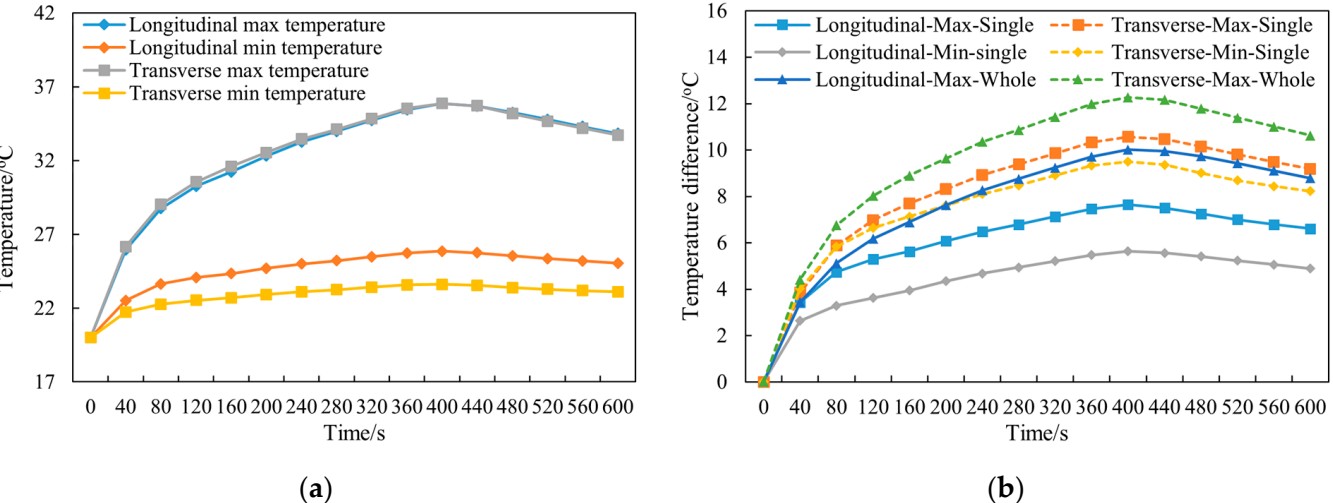

**(a)**                           **(b)**

**Figure 6.** Comparison of longitudinal-flow and transverse-flow simulation based on actual working conditions: (**a**) maximum and minimum temperature profile of the battery pack during discharging; (**b**) the maximum and minimum curves of the maximum temperature difference between the individual cells in the battery pack, and the maximum temperature difference curve of the battery pack as a whole.

### 3.1.2. Effect of Flow Channel Parameters on Thermal Performance of Longitudinal Flow Battery Pack

During the Comsol simulation process, we analyze the impact of the quantity, dimensions, and placement of entry and exit points on the battery pack's cooling. In Figure 7, the curves for the battery pack's maximum temperature and maximum temperature difference are displayed as the number of entrance and exit pairs increases, while keeping the mass flow rate and other conditions consistent. It is evident that a pair of entrances and exits provides the most effective cooling. In Figures 8 and 9, the effect of staggered entrances and exits on battery pack heat dissipation is compared with aligned distribution. It is evident that the staggered distribution of entrances and exits results in better heat dissipation. Our team also analyzed the impact of entrance and exit size, location, and other factors, but the difference was negligible. To save space, these details will not be listed. Based on the analysis, it has been determined that the most effective coolant flow involves a staggered distribution with one inlet and one outlet positioned at the end face. The exact location of the flow channel is depicted in Figure 10, and 4 mm inlet and outlet apertures should be utilized.

To address feasibility concerns in engineering, the battery can be connected via a wire or through a waterproof connecting plate on the battery's end face to create a battery pack suitable for application. While this may increase the axial size of the battery due to the wire and link plate, the proposed method still provides an overall advantage to the energy density of the battery pack.

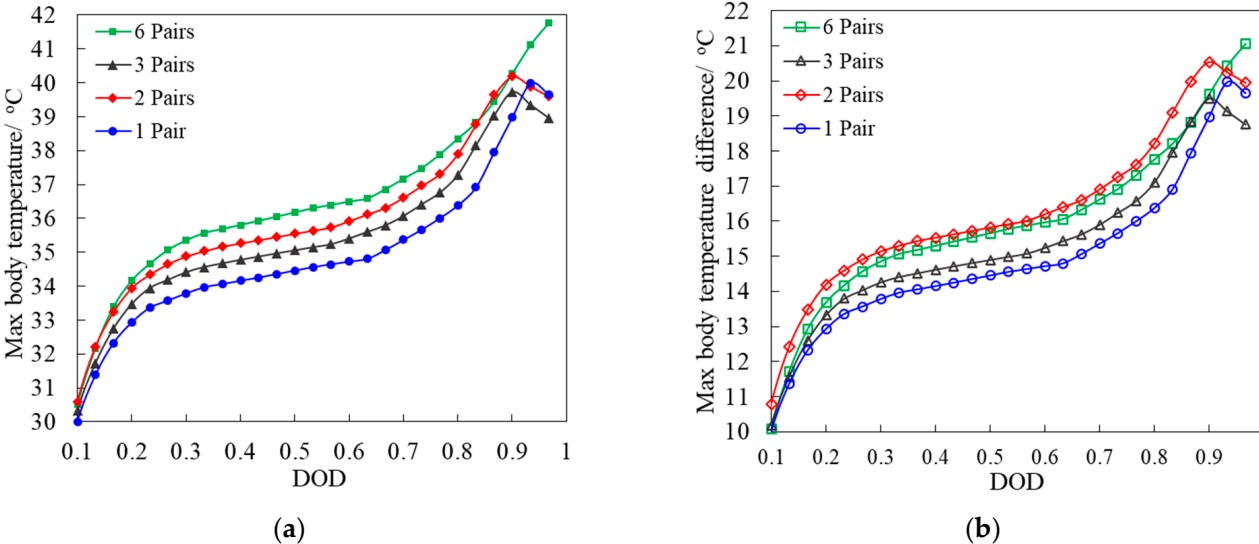

**Figure 7.** Impact of entrance/exit pairs: (**a**) maximum body temperature; (**b**) maximum body temperature difference.

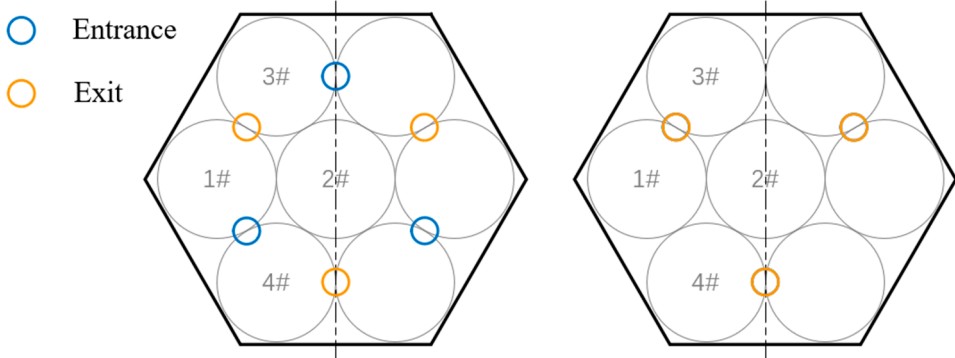

**Figure 8.** Staggered and aligned distribution diagram of entrances and exits.

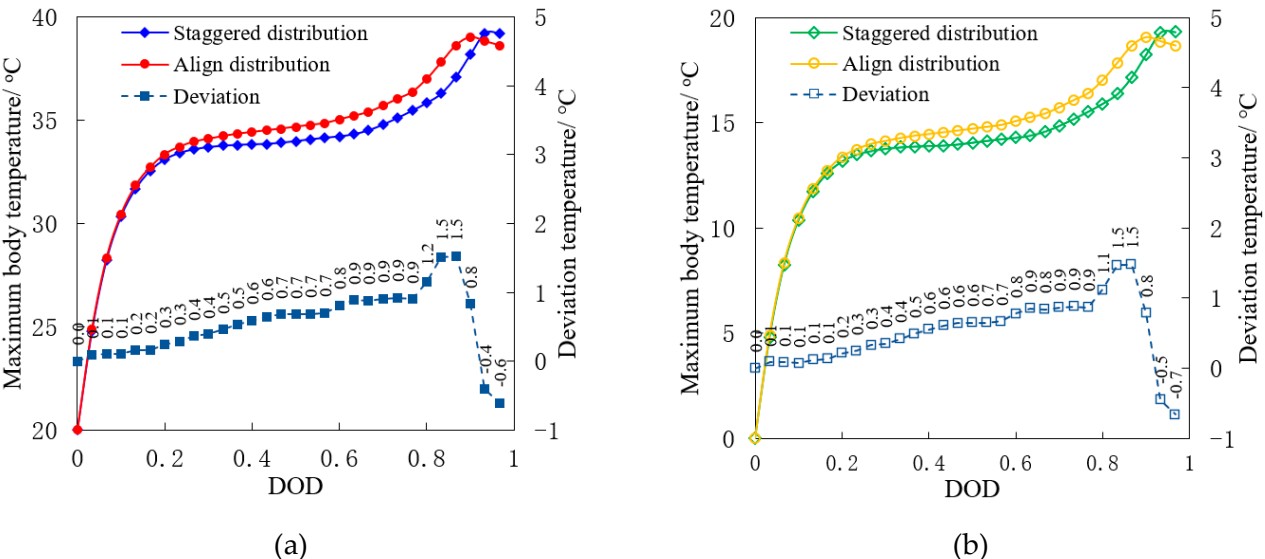

**Figure 9.** Comparison of staggered and aligned distribution of entrances and exits: (**a**) maximum body temperature; (**b**) maximum body temperature difference.

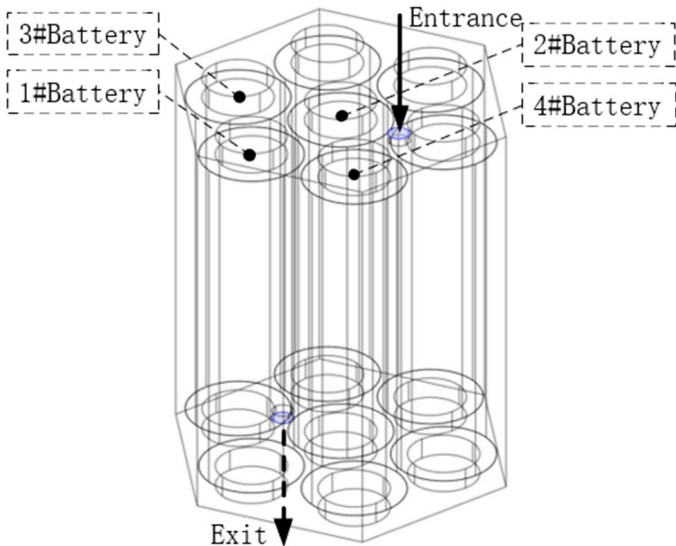

**Figure 10.** Design of flow channel, inlet, and outlet.

### 3.1.3. Effect of Operating Parameters on the Thermal Performance of Longitudinal-Flow Battery Packs

In the previous section, the geometric dimensions and location relationships of the coolant inlet and outlet of the battery pack were studied and selected. In this section, based on this, the effect of the liquid cooling method proposed in this paper on the cooling performance of the battery when the flow rate, discharge rate, and operating conditions are changed is investigated to further verify its feasibility.

(1)    Influence of coolant flow change on cooling effect

The relationship between flow rate and cell temperature is investigated according to the entrance and exit scheme determined in the previous section. The battery is discharged at 4C condition, the coolant flow rate is gradually increased from 0.01 kg/s to 0.19 kg/s, and the change in the maximum temperature and maximum temperature difference of the battery pack is recorded. The simulation results are shown in Figure 11.

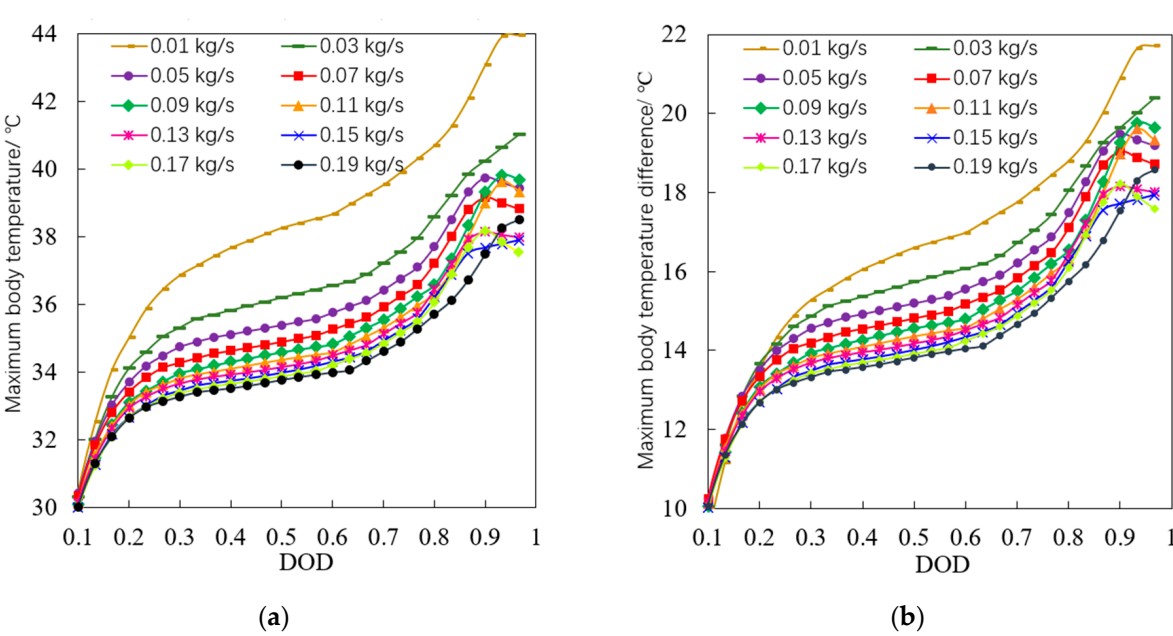

(**a**)                                             (**b**)

**Figure 11.** Body maximum temperature and maximum temperature difference of the battery pack at different flow rates: (**a**) maximum body temperature; (**b**) maximum body temperature difference.

As can be seen from the figures, the maximum body temperature and maximum body temperature difference of the battery pack have approximately the same trend of change throughout the discharge process. In the stage of DOD of 10~85%, the temperature values shown in both figures decrease with the gradual increase in the flow rate, but the magnitude of the decrease is gradually decreasing. There is no such pattern at the stage of DOD of 85% to 100%. During the whole discharge process, the relationship between the extreme value of the maximum body temperature, the maximum body temperature difference, and the flow rate is shown in Figure 12.

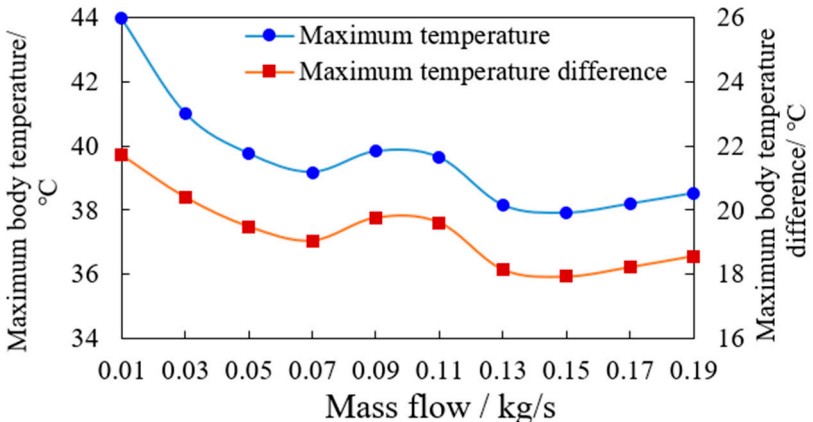

**Figure 12.** Relationship between maximum temperature and maximum temperature difference of the body and coolant flow.

It can be seen from Figure 12 that when the flow rate is 0.01–0.07 kg/s, the maximum body temperature and the maximum body temperature difference of the battery pack decrease rapidly with the increase in the flow rate, and the total decrease can reach 5 °C; In the range of 0.07–0.19 kg/s, the temperature value shows a non-monotonic change with the change in flow rate, which first increases, then decreases, and then increases, and the change range of temperature is only about 1 °C. It can be seen that from 0.07 kg/s, the ability to continuously increase the coolant flow to reduce the maximum temperature and maximum temperature difference of the battery is limited.

(2)   Influence of discharge rate on cooling effect of battery pack

When the flow rate is set to 0.05 kg/s and the discharge rate of the battery is 0.5C, 1C, 2C, 3C, or 4C, the temperature change of the battery pack is studied. The simulation results are shown in Figure 13.

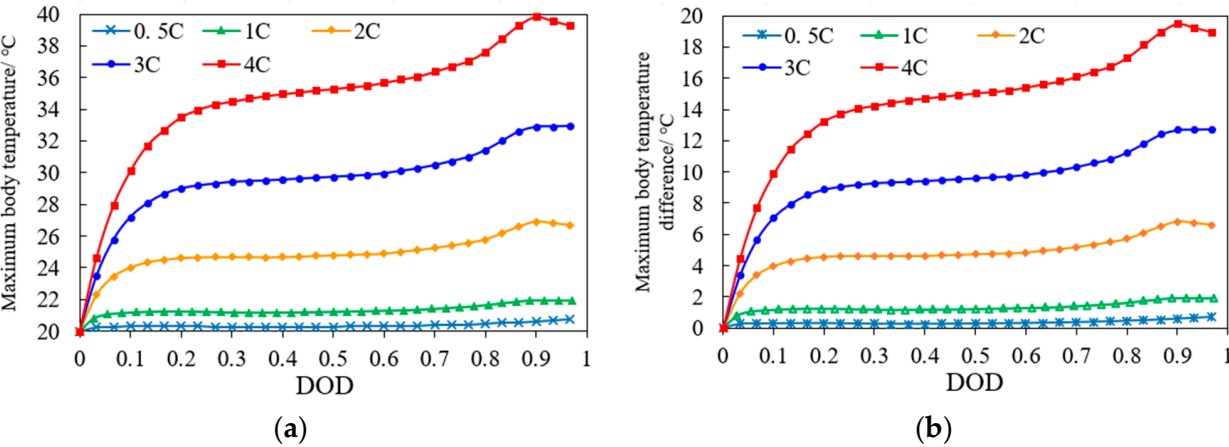

**Figure 13.** The maximum body temperature and maximum temperature difference of the battery pack at different discharge rates: (**a**) maximum body temperature; (**b**) maximum body temperature difference.

It can be seen from the graph that the higher the discharge rate is, the higher the temperature of the battery is. When the battery is continuously discharged at 4C multiplier, its maximum temperature does not exceed 40 °C, and the maximum temperature difference is 19 °C; when it is continuously discharged at 3C rate, its maximum temperature does not exceed 33 °C, and the maximum temperature difference is less than 12 °C. Therefore, the coolant flow rate can be dynamically adjusted according to the operating current and temperature feedback of the battery, using a high flow rate when discharging at high temperature and high current and a low flow rate when discharging at low temperature and low current, in order to reduce the energy consumption of the thermal management system.

3.1.4. Functional Verification in High-Temperature Environment

When in a high temperature environment, the battery and coolant are also affected and become hot. In order to cool the battery down, it is necessary to use refrigeration technology to lower the coolant's temperature to an appropriate level before it can effectively cool the battery. In Figure 14, we can see the curve that represents the changes in the maximum temperature and the maximum temperature difference of the battery pack when the initial temperature of the battery is 40 °C, the initial temperature of the coolant is 20 °C, and the mass flow rate of the cooling water is 0.2 kg/s. Based on the results, we can conclude that the cooling method is effective in addressing the heat dissipation issue of the battery with medium discharge rate even in extreme environmental conditions.

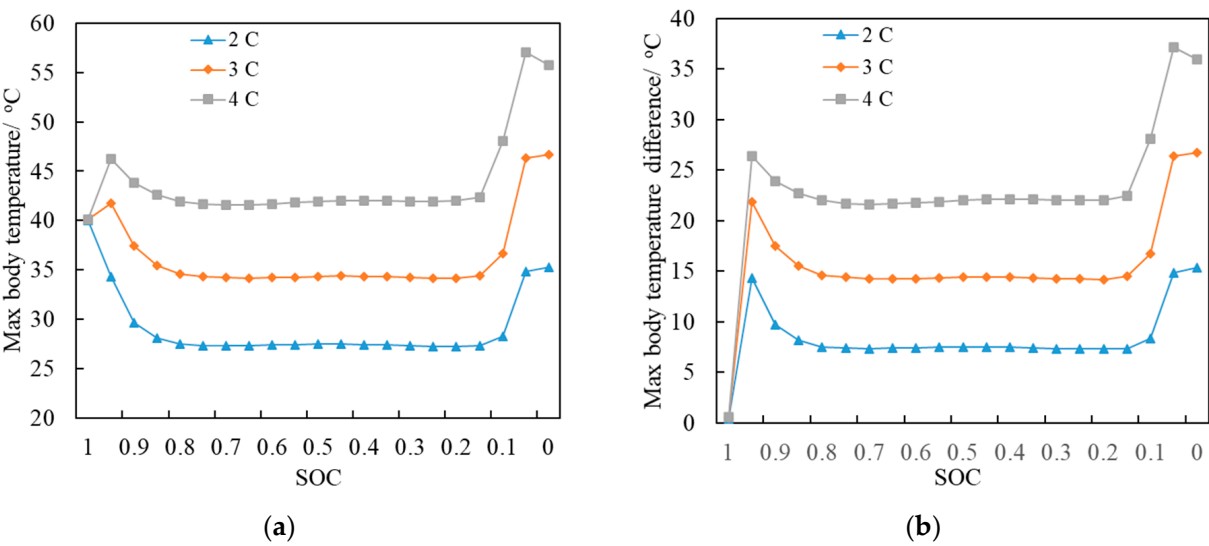

**(a)**　　　　　　　　　　　　　　　　　　**(b)**

**Figure 14.** Temperature change curve when the battery pack is in a high temperature environment: (**a**) maximum body temperature; (**b**) maximum body temperature difference.

*3.2. Analysis of Experimental Results*

3.2.1. Test Results and Analysis under Constant Condition

(1)　The average temperature performance of a battery pack at a high discharge rate

According to the lumped parameter cell model described in Table 2, the simulation is performed under 3C constant current continuous discharge conditions, and the thermal power curve of each cell is recorded as a function of time. This is then converted to the curve of supply voltage versus time for the TDCs according to Ohm's law. The average values of the resistance of the seven TDCs given in Table 4 are taken here for conversion.

This voltage curve is loaded step by the programmable power supply to the seven parallel TDCs. At the same time, the volume flow of cooling water is adjusted to 0.4 L/min (mass flow was 0.0067 kg/s), and the temperature curve is measured as shown in Figure 15.

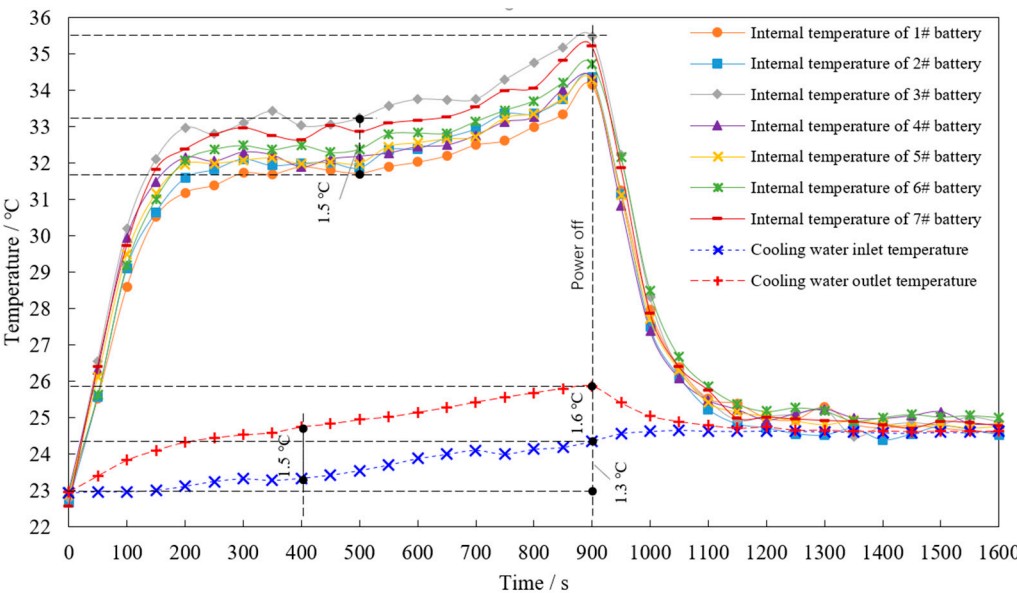

**Figure 15.** Temperature curve of thermal power under 3C discharge condition simulated by TDC with coolant flow of 0.4 L/min.

The test was conducted at room temperature of 23 °C. As can be seen from the Figure 15, the internal temperature of all seven TDCs rose rapidly with the power on, reached the first peak after 200 s, then maintained a steady increase until the second peak at 900 s, and then dropped rapidly with the power off immediately. The cooling water outlet temperature rises slowly as the power is turned on and gradually decreases after the power is turned off. The cooling water inlet temperature also increased slightly during the test, and finally, the cooling water outlet temperature and the internal temperature of the seven TDCs both returned to the same relatively stable value. Due to the self-heating effect of the pump, the cooling water inlet temperature also gradually increased as the test was carried out, and its maximum temperature rise was 1.3 °C. The maximum temperature rise of the outlet cooling water is 2.9 °C, minus the rise in the inlet water temperature, and the maximum value of the temperature difference between the inlet and outlet cooling water is 1.6 °C. It can be converted into heat according to Formula (11).

$$Q = cm\Delta T \tag{11}$$

where Q is the heat carried away by cooling water, in J; c is the specific heat capacity of water, $4.2 \times 10^3$ J/(kg·K); m is the mass of water, in kg; and $\Delta T$ is the temperature difference, in K.

For example, at the 400th second, the calculated heat transfer power is 42.21 W, while the heating power of TDCs is 47.68 W. Without considering other heat dissipation factors such as surface heat transfer, the heating power is greater than the heat dissipation power, and the temperature of the TDCs will continue to rise slowly.

The temperature difference of 7 TDCs in the temperature-holding stage was 1.5 °C. It is observed that the temperature of each TDC has no specific relationship with its distribution in the battery pack. Therefore, the temperature difference should come from two aspects. On the one hand, it comes from the measurement error of the temperature measurement system, including the maximum error of the sensor—about ±0.5 °C. On the other hand, it comes from the manufacturing process of the TDCs and prototype battery pack, including the accuracy of the sensor position inside the TDC and the uniformity of the thermal conductive silicone grease applied between the TDCs and the aluminum tube.

On the whole, under this test condition, the maximum internal temperature rise of the TDCs is 12.8 °C. According to the existing conclusions, the temperature of the central end face of the cell is the highest point of the entire surface temperature and is close to the core temperature of the battery. In order to further observe the temperature distribution of the end face of each TDC, an infrared thermal imager (Fotric 322Pro, Shanghai Thermal Imaging Technology Co, Ltd., Shanghai, China) was used to take photos near the time of highest temperature, as shown in Figure 16. The resolution of the thermal imager is 320 × 240, its temperature measurement accuracy is ±2 °C, and its thermal sensitivity is 0.7 °C. Note that the temperature measured by the infrared thermal imager is the surface temperature, which does not represent the real temperature but only reflects the temperature difference on the surface [30].

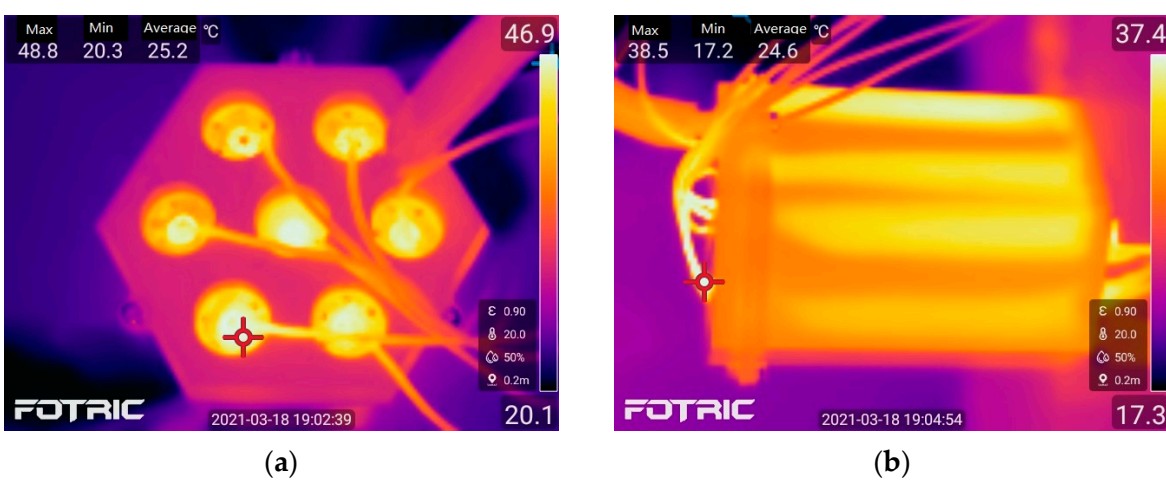

**Figure 16.** Infrared thermal image of the battery pack: (**a**) end face shot; (**b**) side shot.

It can be seen from Figure 16a that there is no obvious distinction between light and dark in the thermal image of the end faces of the seven TDCs, and the temperature is relatively uniform. Consistent with the simulation results, the maximum temperature of the end face of each TDC is located at the center of the end face circle. The highest temperature in this figure reached 48.8 °C. In addition, it can be seen that the wire temperature is quite high, and the closer it is to the TDCs, the higher its temperature is. In Figure 16b, the position of the TDCs in the battery pack is clearly visible. The surface temperature near the corner and edge gap channel is significantly lower than that near the TDCs. In addition, by comparing Figure 16a,b, it can be seen that the average temperature at the side of the battery pack is slightly lower than that at the end.

(2)　Effect of coolant flow rate on heat dissipation performance of longitudinal-flow battery pack

Under the current cooling water flow rate and the thermal power rate of the TDCs, the maximum temperature of TDCs does not exceed 40 °C, but the maximum end temperature exceeds 40 °C. Therefore, it is necessary to further increase the cooling water flow to reduce the temperature and study the relationship between temperature and flow. In order to observe the temperature changes of the seven TDCs more clearly, the average temperature was taken as the research object. Under the same heating power and ambient temperature, the flow of cooling water was changed from 0.2 L/min to 0.2 L/min each time and gradually increased to 0.8 L/min. The relationship between the average temperature and flow of the TDCs is shown in Figure 17.

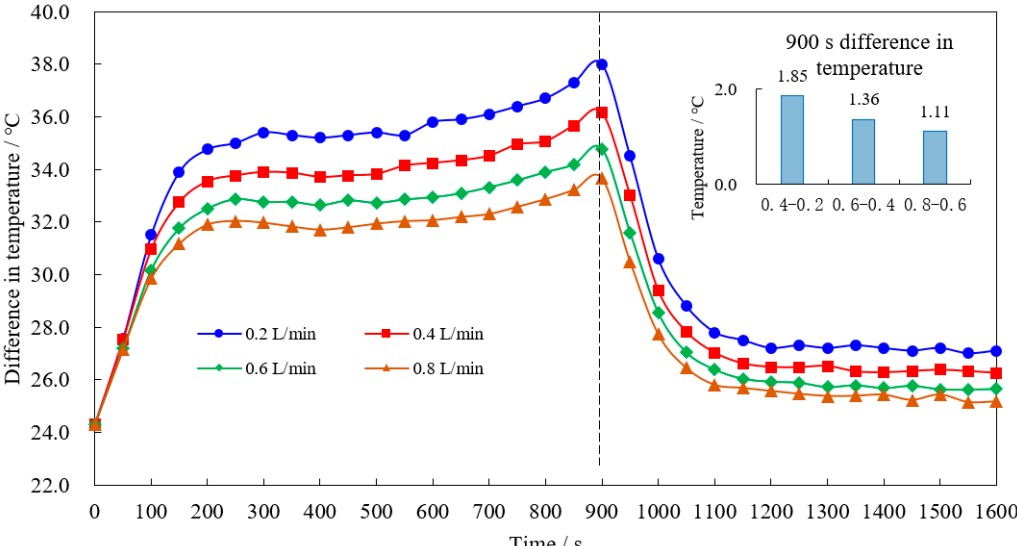

**Figure 17.** Average temperature of 7 TDCs under different cooling water flows.

It can be seen that the average temperature of TDCs is closely related to the coolant flow and decreases with the increase in coolant flow. However, when the coolant flow increases in the sequence of equal difference, the temperature drop gradually decreases. Taking the data in the 900th second in Figure 17 as an example, when the flow rate is gradually increased, the corresponding temperature difference is 1.85 °C, 1.36 °C, and 1.11 °C, and the reduction range is gradually reduced. It can be concluded that there is an upper limit to improving the cooling effect by increasing the flow rate, which is consistent with the above simulation results.

To sum up, the liquid cooling scheme can ensure that the temperature uniformity of each cell in the battery pack is within 1.5 °C, and the maximum temperature inside the TDCs is within 40 °C when the cooling water flow is 0.4 L/min. Furthermore, the maximum temperature can be further reduced by increasing the cooling water flow.

### 3.2.2. Test Results and Analysis under NEDC Conditions

In order to further verify the feasibility of the liquid cooling method proposed in this paper, the NEDC working condition is used for loading in the test bench of this paper. First, in the lumped parameter battery model, the thermal power of the battery is obtained by numerical simulation according to the discharge rate shown in Figure 18 and then converted into the curve of the supply voltage of the TDCs with time according to Ohm's law, as shown in Figure 19. The resistance value used for conversion is the average value of seven TDCs given in Table 4.

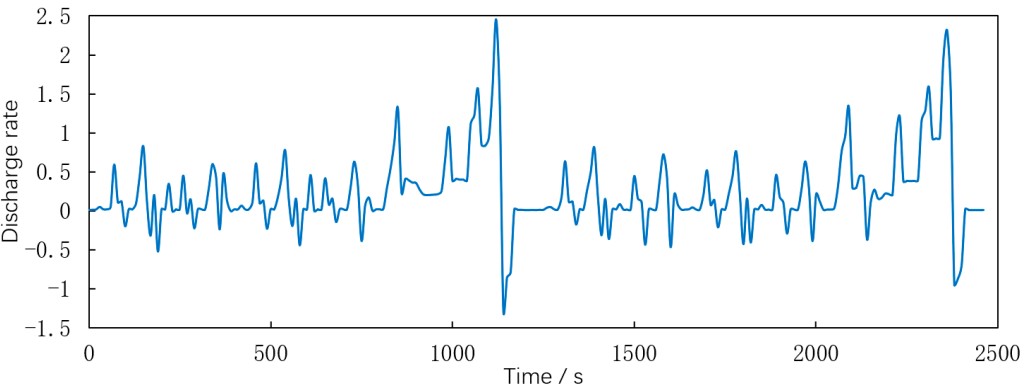

**Figure 18.** Time curve of discharge rate for simulation.

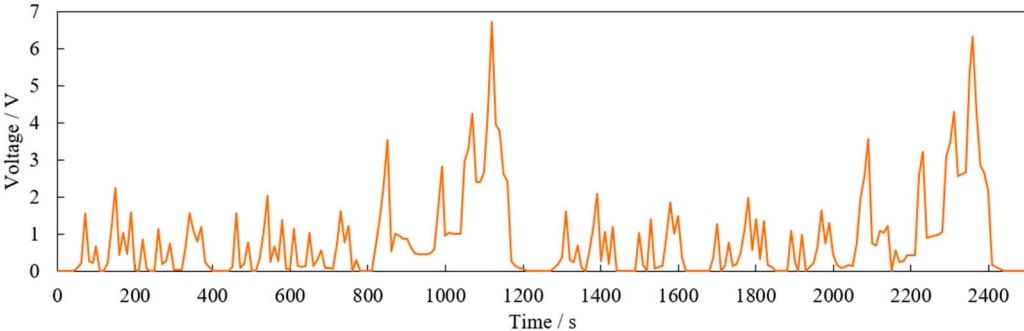

**Figure 19.** Voltage calculated according to NEDC working conditions.

After the curve is discretized in 10 s, it is loaded into the programmable power supply and output to the TDC pack. The total output time is 2500 s, including two NEDC cycle conditions. During the test, the internal temperature of each TDC was measured, and the average value was taken. At the same time, the core temperature of No. 2 TDC was measured and plotted together with the simulation results into the curve shown in Figure 20.

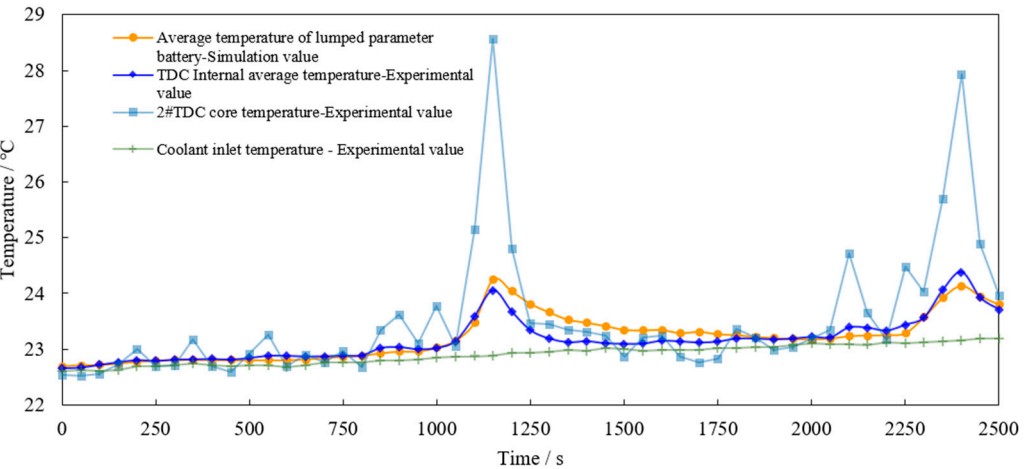

**Figure 20.** Comparison between the measured average temperature of TDCs and the simulated average temperature of lumped parameter battery under NEDC condition.

In the simulation model, the coolant inlet flow of the battery pack is set at 0.2 L/min, which is equal to the flow used in the test. The initial temperature is adjusted to the ambient temperature during the test, 22.6 °C. Due to the self-heating effect of the water pump, there is a basic rising process in the measured data, which is also proved by the slow rise in the coolant inlet temperature. In order to compensate for this error in the simulation model, the inlet water temperature measured in the test is compiled into an interpolation function and used as the inlet temperature of the coolant in the simulation.

Figure 20 shows that under the NEDC working condition, the change trends in the test temperature curve and the simulation curve are basically the same, with two obvious peaks at the same time, and the difference between the maximum temperatures of the two is only 0.2 °C. Within 500 s after the end of the first cycle, the simulation value is slightly higher than the test value. The maximum temperature rise point of the test curve appears 100 s before the end of each cycle, and the temperature rise at its peak is about 1.4 °C. The core temperature curve of the No. 2 battery is also shown in the figure. It can be seen that its response to voltage input is very sensitive, which basically corresponds to the peak of the input voltage curve shown in Figure 18, with the highest peak reaching 28.5 °C. After two NEDC cycles, the internal temperature rise of the TDCs does not exceed 2 °C, the temperature rise of the core temperature is also less than 6 °C, and the peak value of the

second cycle is not higher than that of the first cycle. Therefore, this test shows that the liquid cooling method designed in this paper can effectively cool the battery under NEDC driving conditions.

## 4. Discussion

In the simulation process, the battery lumped parameter model was used for simulation, and in the experimental process, TDCs were used for verification. Although there is a certain error compared with the actual battery, it can be shown by the reference [21] that a TDC battery can meet the needs of this experiment and verified that the longitudinal flow method can perform effective cooling.

The results of above three tests are as follows: (1) The steady-state condition test shows that under the 3C continuous discharge condition and the cooling water flow rate of 0.4 L/min, the temperature uniformity of each TDC in the battery pack is controlled within 1.5 °C, and its internal maximum temperature rise is below 14 °C, decreasing to 10 °C as the flow rate increases to 0.8 L/min. (2) The self-defined condition test shows that under the battery charging and discharging conditions with great dynamic changes, as long as the cooling water flow is maintained at 0.2 L/min, the internal temperature rise of the battery can be controlled at about 5 °C, and the heat dissipation effect is significantly better than natural cooling under the same thermal power. (3) The NEDC working condition test shows that the maximum internal temperature rise of the TDCs is not more than 2 °C and the core temperature rise is only 6 °C under the driving condition close to the actual driving condition.

## 5. Conclusions

This paper focuses on the longitudinal-flow heat dissipation method for cylindrical lithium-ion batteries. The longitudinal-flow heat dissipation method is proposed for cylindrical lithium-ion battery packs. Compared to transverse flow, the average temperature of longitudinal flow is higher, but the maximum temperature is almost the same when considering the actual operating conditions, and the longitudinal-flow technology can significantly reduce the cells' temperature difference and reduce the cells' pack size by about 10%. In order to further verify the feasibility of longitudinal-flow heat dissipation, a study of liquid-cooled longitudinal-flow heat dissipation is conducted in this paper, and numerical simulations conclude that a one-in–one-out coolant flow scheme is the optimal implementation. Further, a multi-channel temperature measurement system and TDCs built in the previous study are used to conduct validation tests of the liquid cooling method. The simulation and tests also proved that the liquid cooling method can restrict the maximum temperature rise in the battery in the battery pack to less than 20 °C under the static condition of continuous discharge of the battery with high current at 3C rate and can restrict the maximum temperature rise to less than 2 °C under the dynamic driving condition of NEDC. The technical solution proposed in this paper not only has sufficient theoretical basis but also has strong practical significance. The liquid cooling method has certain advantages in application scenarios with strict requirements for the arrangement space of battery packs, such as passenger car power batteries and spacecraft power supplies. In addition, this study proposes for the first time a longitudinal-flow heat dissipation method by passing a heat transfer fluid into the gaps formed by the close-packed cylindrical cells.

**Author Contributions:** Conceptualization, W.L.; methodology, W.L. and W.S.; software, W.L.; validation, W.L., W.S., H.H. and G.C.; formal analysis, W.L., W.S. and H.H.; investigation, W.L. and W.S.; resources, W.L. and W.S.; data curation, W.L. and H.H.; writing—original draft preparation, W.L.; writing—review and editing, W.L. and H.H.; visualization, W.L. and H.H.; supervision, W.L.; project administration, W.L.; funding acquisition, S.X. All authors have read and agreed to the published version of the manuscript.

**Funding:** This research was funded by National Key Research and Development Program of China, grant number 2022YFB2502403 and Zhejiang Province "Spearhead" and "Leading Goose" Research and Development Key Program, grant number 2023C01239.

**Data Availability Statement:** Not available.

**Conflicts of Interest:** The authors declare no conflict of interest.

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
