# Peer review of "Study on the Liquid Cooling Method of Longitudinal Flow through Cell Gaps Applied to Cylindrical Close-Packed Battery"

_inventions, doi:10.3390/inventions8040100_

Round 1

Reviewer 1 Report

This paper numerically investigates the battery thermal management using liquid cooling method of longitudinal flow through cell gaps. The topic of the study is interesting and novel. However, there are few points those need to be addressed/justified.

1.      Add one line problem statement at the start of the abstract.

2.      The authors should consider other thermal management techniques for various applications and compare the liquid cooling case with the previous ones. For instance, look into and add the following most recent studies, “Radiative cooling system integrated with heat sink for the thermal management of photovoltaic modules under extreme climate conditions”, “A state-of-the art review on advancing battery thermal management systems for fast-charging”, “PCM-based hybrid thermal management system for photovoltaic modules: A comparative analysis”, “Experimental investigation on the performance of RT-44HC-nickel foam-based heat sinks for thermal management of electronic gadgets”, “Thermal performance analysis of metallic foam-based heat sinks embedded with RT-54HC paraffin: an experimental investigation for electronic cooling

3.      What are the specific thermal properties identified by the genetic algorithm in single cell batteries, and how do they influence the heat dissipation process?

4.      How does the proposed best coolant flow scheme ensure the optimum cooling effect and engineering viability, considering there's only one inlet and one outlet from the end face?

5.      How are Thermal Dummy Cells (TDC) employed to validate the liquid cooling strategy, and how do they represent actual battery cells in terms of thermal behavior?

6.      Given the claim that the longitudinal flow cooling method offers advantages in applications with severe space requirements, how does this method deal with potential challenges of heat buildup in confined spaces?

7.      How does the proposed method deal with extreme scenarios, such as high ambient temperatures or high-discharge operations? Can the maximum temperature rise still be controlled effectively in such conditions?

English language should be thoroughly checked and improved throughout the paper.

Author Response

Question 1: Add one line problem statement at the start of the abstract.

Answer 1: We added this sentence to the beginning of the abstract:

“The popularization trend of electric drive brings opportunities and challenges for the development of lithium battery.”

Question2: The authors should consider other thermal management techniques for various applications and compare the liquid cooling case with the previous ones. For instance, look into and add the following most recent studies, “Radiative cooling system integrated with heat sink for the thermal management of photovoltaic modules under extreme climate conditions”, “A state-of-the art review on advancing battery thermal management systems for fast-charging”, “PCM-based hybrid thermal management system for photovoltaic modules: A comparative analysis”, “Experimental investigation on the performance of RT-44HC-nickel foam-based heat sinks for thermal management of electronic gadgets”, “Thermal performance analysis of metallic foam-based heat sinks embedded with RT-54HC paraffin: an experimental investigation for electronic cooling”

Answer 2: We replaced the description of the cooling method in the original article with this passage :

“As early as the beginning of 2000, Pesaran et al.[10] recognized the importance of thermal management of EV and HEV batteries. After more than two decades of develop-ment, researchers have designed a variety of battery thermal management cooling tech-niques, including gas, liquid, phase change materials(PCM), heat pipes(HP), peltier, and a diverse combination methods.[11] It was also found that, in addition to meet the heat dis-sipation conditions, the battery thermal management system also needs to take into ac-count the vehicle thermal management, battery low-temperature preheating, power con-sumption, volume, temperature uniformity and other design requirements. [12] Although the combination of cooling methods can well realize the cooling function, the optimiza-tion of single cooling methods such as liquid cooling and air cooling is also highly poten-tial. Compared with air cooling, liquid cooling shows obvious advantages and is also the mainstream in commercial applications.[13] PCM and HP are both passive cooling methods, which have been widely noticed because of their outstanding performance.[14] Phase change cooling has many designs in addition to lithium battery cooling, there are also studies in electronic component cooling and photovoltaic panel cooling. [15,16] However, the PCM has a small thermal conductivity, easy to failure after heat accumula-tion, low volume density and other problems. The HP has a high manufacturing cost, in-stallation complexity, difficult to maintain. Although many optimization schemes have been proposed, [17] there is still a long way from practical commercial applications. [14] Liquid cooling is now the most popular battery thermal management solution, taking into account the technical viability, heat dissipation effect, and economic cost.”

Question3: What are the specific thermal properties identified by the genetic algorithm in single cell batteries, and how do they influence the heat dissipation process?

Answer 3: We obtained the thermal conductivity of the cell in axial and radial directions as well as the convective heat transfer coefficient on the cell surface by a genetic algorithm. And we added 2.1.3 Genetic Algorithm in the paper to describe the application process of the genetic algorithm. The detailed description is as follows:

“Because the thermal conductivity of cells in axial and radial directions are different, it is necessary to use the thermal conductivity coefficients and convective heat transfer coefficients of the actual cells in the simulation process of comsol. In order to obtain more accurate thermal coefficients as mentioned above, the measured single cell discharge temperature is taken as output, entropy coefficients, cell terminal voltage and open-circuit voltage are taken as inputs, and the thermophysical parameters of the single cell are recognized by genetic algorithm. In the simulation process of the thesis, the battery lumped parameter model is used, and the battery is regarded as a uniform heat-generating object. The heat is generated inside the battery and conducted outward, finally transferred to the coolant through the surface of the battery, and brought out of the battery pack by the coolant. In this process, assuming the battery heat generation power is accurate, the battery thermal conductivity and surface heat transfer coefficient directly determine the accuracy of the simulation results.”

Question4: How does the proposed best coolant flow scheme ensure the optimum cooling effect and engineering viability, considering there's only one inlet and one outlet from the end face?

Answer 4: In our resubmission of the manuscript, we added the content of the question you asked in section 3.1.2 Effect of flow channel parameters on thermal performance of longitudinal flow battery pack, as described below:

“In comsol simulation process we compare the effects of the number, size and location of the entrances and exits on the cooling process of the battery pack, respectively. Figure 7 show the curves of the maximum temperature and the maximum temperature difference of the battery pack when the number of pairs of entrances and exits is increased while keeping the mass flow rate and other conditions are the same, and it can be seen that a pair of entrances and exits has the best cooling effect. Figure 8 and Figure 9 compare the impact of staggered entrances and exits on the heat dissipation of the battery pack, it can also be clearly illustrated when the entrances and exits staggered distribution heat dissi-pation effect is better, in addition to the following results, our team also analyzed the im-pact of the entrance and exit size, location and other factors, the difference is small, in or-der to save space will not be listed in detail. It is concluded that the optimal coolant flow is staggered distribution of one inlet and one outlet from the end direction. The specific loca-tion of the flow channel is shown in Figure 7, and the 4mm inlet and outlet apertures are used.

For engineering feasibility issues, the battery can be connected directly through the wire, also can be achieved through the addition of waterproof connecting plate in the bat-tery end face to form a battery pack for application, the wire and the link plate will in-crease the axial size of the battery, but on the whole, the cooling method of the overall en-ergy density of the battery pack is still advantageous.

Question5: How are Thermal Dummy Cells (TDC) employed to validate the liquid cooling strategy, and how do they represent actual battery cells in terms of thermal behavior?

Answer 5: We added the following description at the beginning of section 2.2.1 TDC-based pack principle

In our team's earlier publications, [27] we verified the feasibility of developing a bat-tery thermal management system(BTMS) using TDC instead of a real battery. Figure 3 and Figure 4 of this thesis introduce the physical structure of the TDC battery and the assem-bly structure of the experiments with the TDC battery, respectively. The essence of TDC simulation of the actual battery is to directly heat the coiling filler through the electric heating wire, so that the TDC has the same or even bigger thermal power than the battery, and the TDC has similar axial and radial thermal conductivity and convective heat trans-fer coefficient with the actual battery, so by measuring the thermal power of the battery under the corresponding charging and discharging rate in advance, the battery hermal behavior can be simulated by the TDC. Based on this working principle, the TDC cell can simulate the thermal phenomena of a wide rate of battery operation, even in the case of thermal runaway, in a much safer and more reliable way.

Question6: Given the claim that the longitudinal flow cooling method offers advantages in applications with severe space requirements, how does this method deal with potential challenges of heat buildup in confined spaces?

Answer 6:We have reviewed some literature believes that the heat buildup should be the result of the long-term work, and the heat dissipation is not timely in some parts of the battery pack to form a heat accumulation area, making the local temperature is too high to cause damage or even exist the risk of thermal runaway. However, we simulated the thermal performance of the entire process of high rate discharge of the battery or battery pack. It should be noted that the heat generation power of the battery charging is lower than that of the discharge, the main working condition of the BTMS is the battery discharging process. And we added the simulation cooling process of this cooling method in high temperature environment in the revision content of the paper. We feel that the simulation results are able to show that the cooling method is able to cool continuously charged and discharged batteries and overcome the problem of heat buildup in batteries.

Question 7: How does the proposed method deal with extreme scenarios, such as high ambient temperatures or high-discharge operations? Can the maximum temperature rise still be controlled effectively in such conditions?

Answer 7: We have added section 3.1.4 Functional verification at high temperature environment to the paper to validate the performance of the cooling method in extreme environments. The specific descriptions are as follows:

“At high temperature environment, both the battery and the coolant are in a high temperature state, in order to realize the cooling of the battery, it is necessary to reduce the coolant to a suitable temperature by refrigeration technology, and then cool the battery. Figure1 shows the change curve of the maximum temperature and the maximum temperature difference of the battery pack when the initial temperature of the battery is 40oC, the initial temperature of the coolant is 20 oC, and the mass flow rate of the cooling water is 0.2 kg/s. The results show that the cooling method can still solve the heat dissipation problem of the battery with medium discharge rate under extreme environment.

Reviewer 2 Report

The paper deals with a liquid cooling method for cylindrical close-pack batteries. The simulation is fairly presented, but it should be extended (at least theoretically) for other types of cooling agents. As long as the main purpose is related to large batteries for transport domain, the usual liquids already in use for transport should be simulated, e.g. antifreeze. On the other hand, it is obvious that the cooling flow could be related to a hybrid system for vehicle cooling, to perform both functions, so more parameters of liquid flow should be mentioned, in line with the ones normally related to vehicle cooling system. The simulation is made for a package of cylindrical batteries, but at least a theoretical extension for prismatic batteries should be mentioned, as long as Li batteries are mentioned as study target. In all, the experimental part is comprehensive. The conclusions must be extended, to include more specific technical conclusions related to the comparison of simulation-experiment, and also address the possibilities of the extension of the model for other types of liquids and batteries. 

Some terms and also some spelling items must be revised.

Author Response

To explain our choice of water as a coolant, we have added the following to section 2.1.5 Initial and boundary conditions. The specific descriptions are as follows:

“Water as a coolant is widely used in many fields, including machinery, electrical products and internal combustion engines, and the most commonly used coolants in BTMS are also water and ethylene glycol water solution. [13,24] In addition to water and ethylene glycol water solution, the coolants applied in BTMS also include oil, nanofluid, and liquid metal, and the characteristics of the use of the above coolants are summarized in the paper [25], where nanofluid and liquid metal are the coolants that are superior in heat dissipation performance relative to water in numerical simulation, and oil is often used as a coolant for the direct cooling method, which has a unique advantage, and the ethylene glycol water solution reduces the cooling ability of water relative to water, but the degree of reduction is related to the larger concentration of ethylene glycol in mass, the mass concentration of 50% ethylene glycol is the most commonly used antifreeze, but the paper [26] compared the cooling effect of ethylene glycol water solution and pure water, the maximum difference in temperature is less than 2 ℃, and the maximum difference in the temperature difference is less than 1 ℃, so the water as the coolant for the validation of cooling method in this thesis is reasonable.”

But our longitudinal flow cooling scheme is derived with the help of the gap that exists when the cylindrical lithium batteries are densely arranged as the flow channel, and the prismatic batteries do not have this feature, so it is not discussed. Cylindrical and diamond-shaped batteries have their own characteristics, but as far as relevant reports show, large-size cylindrical batteries are very promising, from 18650 to 21700 batteries or Tesla 4680 batteries, the development trend of large-size cylindrical batteries is obvious, so our team's design is meaningful for the development of batteries.

Round 2

Reviewer 2 Report

The authors fairly responded to the observations of the reviewers.